# Low Switching Loss Built-In Diode of High-Voltage RC-IGBT with Shortened P+ Emitter

**DOI:** 10.3390/mi14040873

**Published:** 2023-04-19

**Authors:** Wei Wu, Yansong Li, Mingkang Yu, Chongbing Gao, Yulu Shu, Yong Chen

**Affiliations:** School of Automation Engineering, University of Electronic Science and Technology of China, Chengdu 611731, China

**Keywords:** RC-IGBT, reverse recovery characteristics, IGBT characteristics

## Abstract

In this paper, a low switching loss built-in diode of a high-voltage reverse-conducting insulated gate bipolar transistor (RC-IGBT) is proposed without deteriorating IGBT characteristics. It features a particular shortened P+ emitter (SE) in the diode part of RC-IGBT. Firstly, the shortened P+ emitter in the diode part can suppress the hole injection efficiency resulting in the reduced carriers extracted during the reverse recovery process. The peak of the reverse recovery current and switching loss of the built-in diode during reverse recovery is therefore lowered. Simulation results indicate that the diode’s reverse recovery loss of the proposed RC-IGBT is lowered by 20% compared with that of the conventional RC-IGBT. Secondly, the separate design of the P+ emitter prevents the performance of IGBT from deteriorating. Finally, the wafer process of the proposed RC-IGBT is almost the same as that of conventional RC-IGBT, which makes it a promising candidate for manufacturing.

## 1. Introduction

To minimize the size of IGBT modules and manufacturing costs meanwhile improving the power conversion efficiency, the reverse-conducting insulated gate bipolar transistor (RC-IGBT) has been proposed [1,2,3]. An IGBT and a freewheeling diode are integrated into one chip which enables RC-IGBT both during forward and reverse conduction [4,5,6,7]. Due to the advantages above, RC-IGBT is a promising device in the power semiconductor field. However, RC-IGBT also suffers from some inherent defects. The snap-back occurring in the forward conduction state is a well-known problem [8,9]. Furthermore, RC-IGBT has complex trade-off relationships between IGBT characteristics, IGBT’s on-state forward voltage drop (*V_CE_*_(*sat*)_) and the turn-off energy loss (*E_off_*), and built-in diode characteristics, diode’s forward voltage drop (*V_on,diode_*) and the diode’s reverse recovery loss (*E_rr_*). Therefore, it is difficult to obtain a well-balanced performance between IGBT and a built-in diode. The *E_rr_* is the main proportion of switching energy loss for RC-IGBT. Lifetime control techniques [10,11] are widely used for a diode in the case of a high-frequency operation. However, the heavy lifetime control realizing the low diode switching energy loss will severely sacrifice the IGBT’s performance.

There are many efforts to solve this problem [12,13]. In recent years, a new contact structure of RC-IGBT is also proposed to lower the diode switching loss by reducing the injection efficiency [2]. In summary, there are two methods to improve the trade-off relationship between *V_on,diode_* and *E_rr_* for RC-IGBT. One is changing the carrier lifetime in Si bulk by the electron beam or proton irradiation to control the carrier transport efficiency; the other is changing the anode or cathode structure to control the carrier injection efficiency.

In this paper, a low *E_rr_* diode of RC-IGBT with a particular shortened P+ emitter (SE RC-IGBT) is proposed, which is almost not changing the wafer process of the conventional RC-IGBT. The peak of reverse recovery current density (*J_PR_*) and switching loss of the built-in diode during reverse recovery are reduced by shortening the length of the P+ emitter in the diode part. Simulation results show that the *J_PR_* is lowered by 17.7%, from 16.6 A/cm^2^ to 14.1 A/cm^2^, while the IGBT characteristics are maintained.

## 2. Device Structure and Mechanism

Figure 1 shows the cross-sections of conventional RC-IGBT (*Con.* RC-IGBT) and SE RC-IGBT. There is a carrier stored (CS) layer below the P-base to reduce *V_CE_*_(*sat*)_. The dummy trenches used to reduce the electric field in the carrier stored layer increase the breakdown voltage. Comparing with the *Con*. RC-IGBT, the proposed SE RC-IGBT features particularly shortened P+ emitters, which are set beside the dummy trench. Further, the P+ emitters between regular trench gates are the same as the *Con*. RC-IGBT’s.

The turn-off current density characteristics and an approximate representation of carrier distribution at different time intervals during the reverse recovery of the built-in diode are shown in Figure 2a,b, respectively. The *J_PR_* is the reverse peak recovery current density at time *t_p_* illustrated in Figure 2a, when the built-in diode begins to support the reverse voltage. At this time, the carrier density at *J*_1_ junction becomes zero, as illustrated in Figure 2b. The switching time *t_rr_* is defined as the time interval between *t*_0_ and the time when the reverse recovery current has decayed down to 20% of *J_PR_*. The subdivision of *t_rr_* in *t_a_* and *t_b_* is shown in Figure 2a. It is assumed that the carrier density decreases at *J*_1_ junction while remaining relatively constant *n_c_* within the N-drift region at a distance b.

Under high-level injection, the on-state current density (*J_F_*) at *t* = 0 can be written as [14],
(1)$$J_{F}=-\frac{1}{2qD_{P}}\cdot \frac{dn(x)}{dx}|{}_{x=0}$$
where *D_p_* is the bipolar diffusion constant, *q* is the charge constant. It can be obtained from Figure 2b,
(2)$$\frac{dn(x)}{dx}|{}_{x=0}=\frac{n_{c}-n(0)}{b}$$

Substituting Equation (2) into Equation (1), *b* can be given by,
(3)$$b=\frac{2qD_{P}[n(0)-n_{c}]}{J_{F}}$$

Under the assumption that the carrier density varies linearly over the distance *b* from 0 to *n_c_*, the *J_PR_* at time *t_p_* is given by,
(4)$$J_{PR}=-2qD_{P}\frac{n_{c}}{b}$$

Making the assumption that the on-state current density is determined by recombination, the *J_F_* can also be expressed as follow,
(5)$$J_{F}=\frac{Q}{\tau_{HL}}=\frac{qn_{c}W}{\tau_{HL}}$$
where *Q* is the stored charge in the N-drift region due to the on-state current flow, *τ_HL_* is the high-level lifetime in the N-drift region, and *W* is the length of the N-drift region.

Substituting Equation (5) into Equation (4)
(6)$$J_{PR}=\frac{\tau_{HL}J_{F}{}^{2}}{qW[n_{c}-n(0)]}$$

The *n*(0) is given by [14]
(7)$$n(0)=p(0)=N_{A}\exp(\frac{qV}{2k_{0}T})$$
where *V* is the voltage drop at *J*_1_ junction, *k*_0_*T* is the temperature constant, *N_A_* is the doping concentration of the P-type region in the built-in diode, including the P-base region and the P+ emitter region. Substituting Equation (7) into Equation (6), *J_PR_* can be written as
(8)$$J_{PR}=\frac{\tau_{HL}J_{F}{}^{2}}{qW[n_{c}-N_{A}\exp(\frac{qV}{2k_{0}T})]}$$

The *N_A_* of SE RC-IGBT decreases by shortening the length of the P+ emitter in the diode part. Further, *n_c_* remains due to the *J_F_* being unchanged according to Equation (5). Therefore, the *J_PR_* of SE RC-IGBT is supposed to be decreased by Equation (8).

Figure 3 shows the hole density distribution when the built-in diode is turn-on which is the same as the electron density distribution due to the principle of electric neutrality. It is seen that the hole density of SE RC-IGBT is evidently lower than that of the *Con.* RC-IGBT in the drift region closed to the emitter due to a lower carrier injection efficiency of the shortened P+ emitter. Thus, the *J_PR_* of SE RC-IGBT must be lower than that of *Con.* RC-IGBT, according to Equation (6). Reducing *J_PR_* is an effective method to lower the diode switching loss [15]. In addition, thanks to the separate design of the P+ emitters in SE RC-IGBT, hole injection efficiency in the IGBT part is not affected, which prevents the IGBT characteristics from deteriorating. Consequently, the proposed SE RC-IGBT can lower the *E_rr_* without sacrificing IGBT performance and changing the wafer process.

## 3. Simulation and Discussions

In order to verify the characteristics of the proposed SE RC-IGBT, numerical simulations were performed by TCAD SENTAURUS [16]. A breakdown voltage of 1.2 kV and a threshold voltage of 5 V was designed. The related structure parameters are listed in Table 1. The area of the active region of the SE RC-IGBT is set to 1 cm^2^.

Figure 4 shows the impact of P-collector width *w_Pc_* on the snap-back voltage (*V_SB_*) of SE RC-IGBT. It can be seen that the snap-back effect was suppressed as the increase in *w_Pc_*, which was almost eliminated when w_Pc_ increased to 240 μm. The length of the P-collector was set to 240 μm. Therefore, the ratio between the length of the P-collector and the length of the N-collector was set to 8:1.

Figure 5 shows the simulated test circuit for investigating the turn-off behavior of the built-in diode with inductive load. The inductance in the main circuit is set to 200 μH. The reverse recovery current density (*J_RR_*) waveforms of the built-in diodes for SE RC-IGBT and *Con.* RC-IGBT is shown in Figure 6. It can be seen that the *J_PR_* and the collector voltage overshoot of SE RC-IGBT are obviously lower than those of *Con.* RC-IGBT during the reverse recovery. Furthermore, there is a positive correlation between the *J_PR_* and the width of the P+ emitter *w_p+_*.

Figure 7 shows the dependence of the *J_PR_* and softness factor *S* on the width of the P+ emitter *w_p+_*. It can be seen that the *J_PR_* of SE RC-IGBT is lowering with the decrease in *w_p+_*, which is consistent with the derivation of Equation (8). Furthermore, the *J_PR_* of SE RC-IGBT is decreased by 17.7% when *w_p+_* = 0.5 μm, from 16.6 A/cm^2^ to 14.1 A/cm^2^, compared with that of the conventional RC-IGBT. The softness factor *S* is defined as follows:(9)$$S=\frac{t_{b}}{t_{a}}$$
where *t_b_* and *t_a_* are mentioned in Figure 2a. The softness factor represents the attenuation rate of the reverse recovery current. The larger *S* causes the smaller voltage spike, which means a more secure system. It can be seen that the softness factor *S* is almost unchanged with the decrease in the P+ emitter width.

Figure 8 shows the dependence of on-state voltage (*V_on,diode_*) and *E_rr_* of the built-in diode on the width of the P+ emitter. The *V_on,diode_* increases with the decrease in the P+ emitter width due to the suppression of the carrier injection efficiency of the P+ emitter. However, the *E_rr_* of SE RC-IGBT is decreased by 19% when the P+ emitter width is shortened from 5 μm to 0.5 μm.

Figure 9 shows the inductive load circuit for the switching simulation of RC-IGBT. The inductive load is set to 200 μH. The IGBT turn-off characteristics of *Con.* RC-IGBT and SE RC-IGBT are shown in Figure 10. It can be seen that the turn-off time of SE RC-IGBT is almost the same as that of the conventional RC-IGBT.

Figure 11 shows the dependences of breakdown voltage (BV) on *w_p+_* and the trade-off relationship between *E_off_* and *V_CE_*_(*sat*)_ at different P-collector doping concentrations for Con. RC-IGBT and SE RC-IGBT. It is seen that the BV of SE RC-IGBT is the same as that of the conventional RC-IGBT because the IGBT part is unchanged. Furthermore, the trade-off relationship between *E_off_* and *V_CE_*_(*sat*)_ of SE RC-IGBT is also not deteriorated. As a result, the SE RC-IGBT improves the turn-off characteristics of the built-in diode without deterioration of IGBT performance.

The simulated temperature, collector–emitter voltage, and collector current waveforms of the Conv. RC-IGBT and SE RC-IGBT with a 0.5 ms unclamped inductive switching (UIS) pulse are shown in Figure 12. The on-stage time *t_ON_* is 0.5 ms to make the avalanche current *I_av_* reach 50 A for both Conv. RC-IGBT and SE RC-IGBT. The maximum collector voltages of SE RC-IGBT were reduced by 14% compared to that of Conv. RC-IGBT from 1409.4 V to 1200.3 V. The avalanche time *t_av_* of SE RC-IGBT was increased by 19% compared to that of Conv. RC-IGBT from 46 μs to 55 μs. So the d*I_av_*/d*t* of SE RC-IGBT is 0.9 A∙μs^−1^, which was reduced by 18% compared to 1.1 A∙μs^−1^ of Conv. RC-IGBT. Furthermore, the maximum temperature during the UIS transient in SE RC-IGBT reached 530.9 K lower than 693.7 K in the Conv. RC-IGBT, which indicates that the UIS failure in the SE RC-IGBT will be less likely to be triggered by a high-temperature transient.

Figure 13 shows the simple key processing steps to fabricate the front side of the SE RC-IGBT. The backside processing steps are not needed to be addressed in this article. Instead of changing the front-side processing, the modification of the lithographic mask is only required before the P+ emitter ion implant. First, grow oxide to suppress the channeling effect in the ion implantation, as shown in Figure 13a. Next, redefine the lithographic mask and BF_2_ ion implant, as shown in Figure 13b. Furthermore, N+ collectors are formed by phosphorus ion implantation, as shown in Figure 13c. Then, anneal and form the metal electrode, as shown in Figure 13d. Figure 14 shows the process flow for SE RC-IGBT. The wafer process of SE RC-IGBT is the same as that of *Con*. RC-IGBT. Only the lithographic mask needs to be redefined at the step of the P+ emitter ion-implant circled by dotted lines.

The *E_rr_* of different kinds of RC-IGBT are compared in Figure 15. The RC-IGBT with the new contact structure in [2] shows the lowest *E_rr_*. However, it has a difficult and highly cost manufacturing. Lifetime control by electron beam irradiation in [10] not only changes the built-in diode characteristics but also deteriorates the IGBT performance. Although the SE RC-IGBT proposed in this paper shows an inferior effect of decreasing *E_rr_*, it does not change the wafer process of conventional RC-IGBT and sacrifice the IGBT characteristics.

## 4. Conclusions

A particular shortened P+ emitter RC-IGBT (SE RC-IGBT) is proposed and investigated by simulation. The characteristics of IGBT and the built-in diode are both discussed in this paper. Simulation results show that the *E_rr_* of SE RC-IGBT has been decreased by about 20% compared with that of the conventional RC-IGBT. Furthermore, the soft factor *S* is almost unchanged. In addition, the SE RC-IGBT process is compatible with the conventional RC-IGBT process, making it a promising candidate for production.

## Figures and Tables

**Figure 1 micromachines-14-00873-f001:**
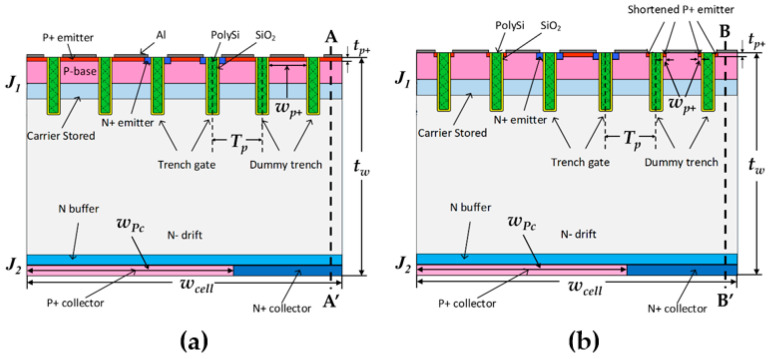
The cross-section of RC-IGBT. (**a**) Conventional RC-IGBT; (**b**) The proposed SE RC-IGBT.

**Figure 2 micromachines-14-00873-f002:**
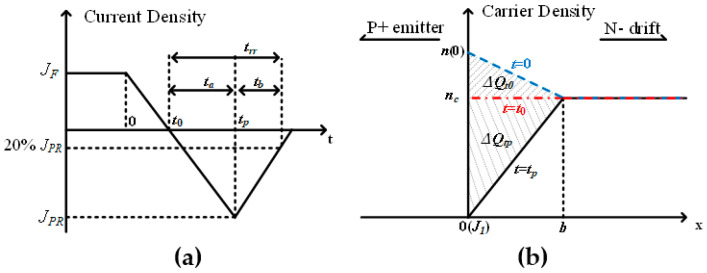
The model of the built-in diode. (**a**) The linearized reverse recovery current density waveform of the built-in diode; (**b**) The change in carrier distribution at different time intervals during the turn-off process of the built-in diode.

**Figure 3 micromachines-14-00873-f003:**
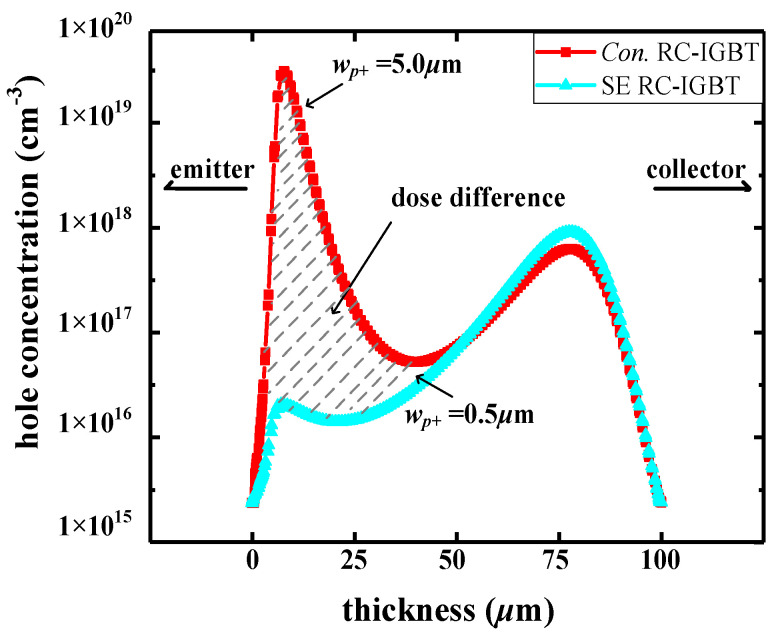
Hole density distribution of Conventional RC-IGBT along line AA’ and SE RC-IGBT along line BB’.

**Figure 4 micromachines-14-00873-f004:**
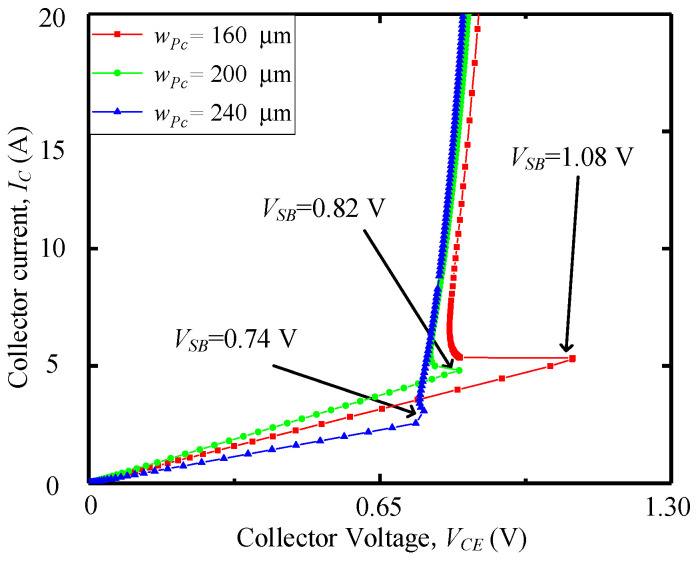
Impact of the P-collector width *w_Pc_* on snap-back.

**Figure 5 micromachines-14-00873-f005:**
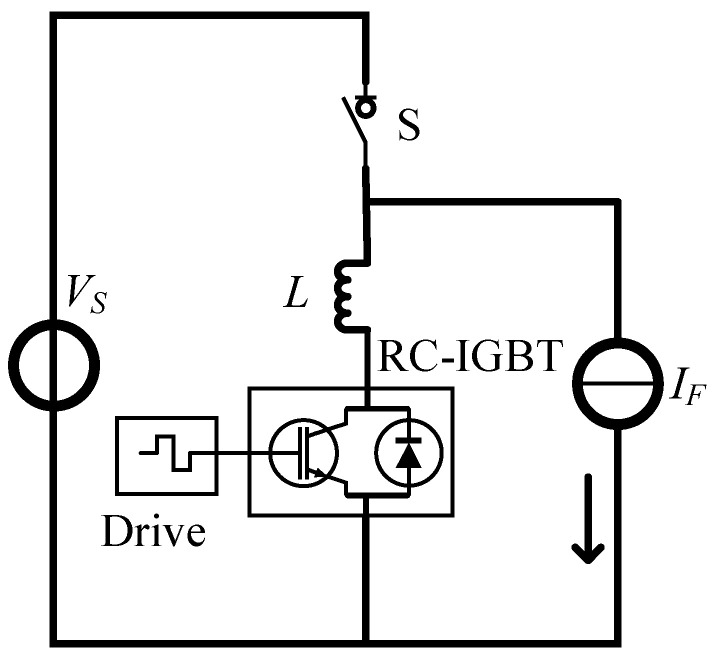
Inductive load test circuit for turn-off behavior of built-in diode.

**Figure 6 micromachines-14-00873-f006:**
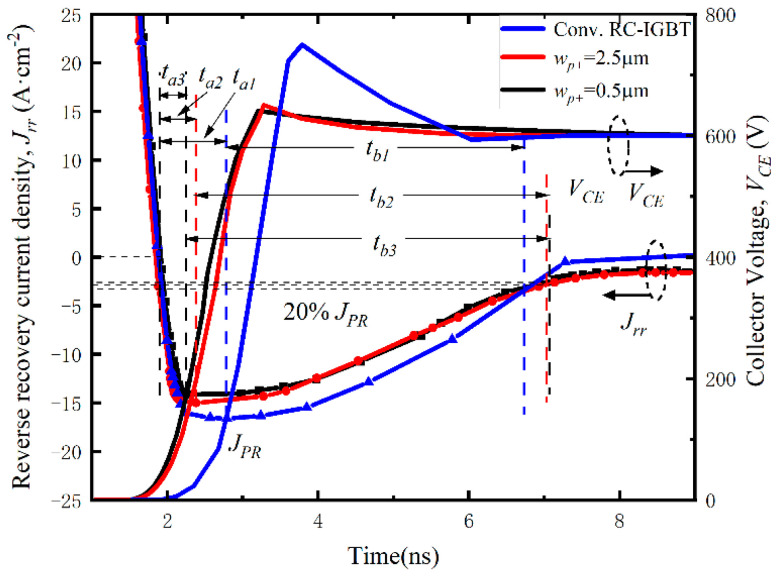
Reverse recovery current and collector voltage waveforms of conventional RC-IGBT and SE RC-IGBT. (*t_a_* and *t_b_* are mentioned in Figure 2a).

**Figure 7 micromachines-14-00873-f007:**
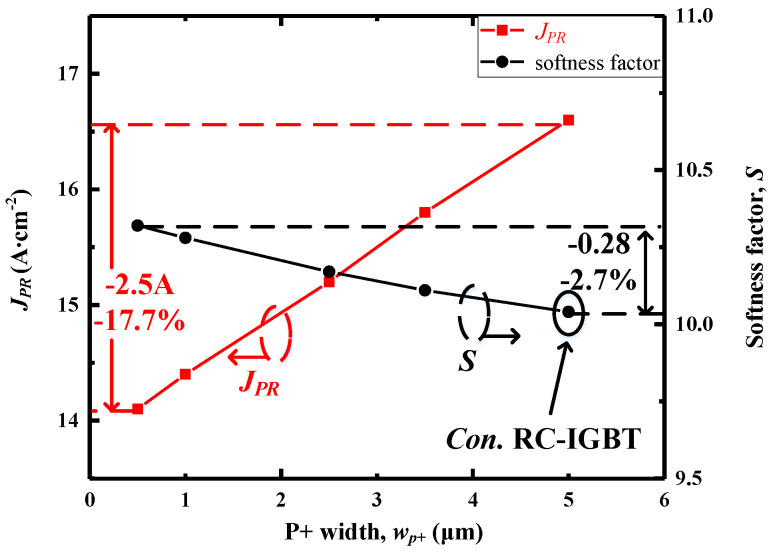
The change in the *J_PR_* and softness factor *S* with different P+ emitter widths.

**Figure 8 micromachines-14-00873-f008:**
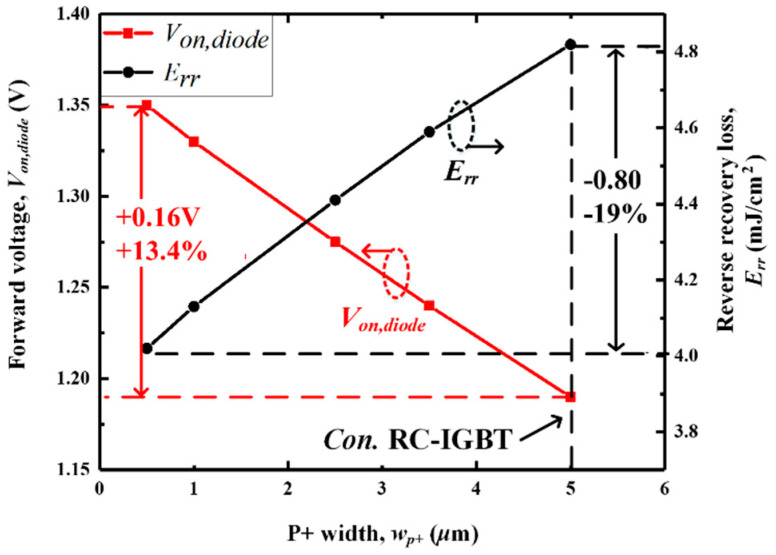
The change of the *V_on.,diode_* and *E_rr_* with different P+ emitter widths.

**Figure 9 micromachines-14-00873-f009:**
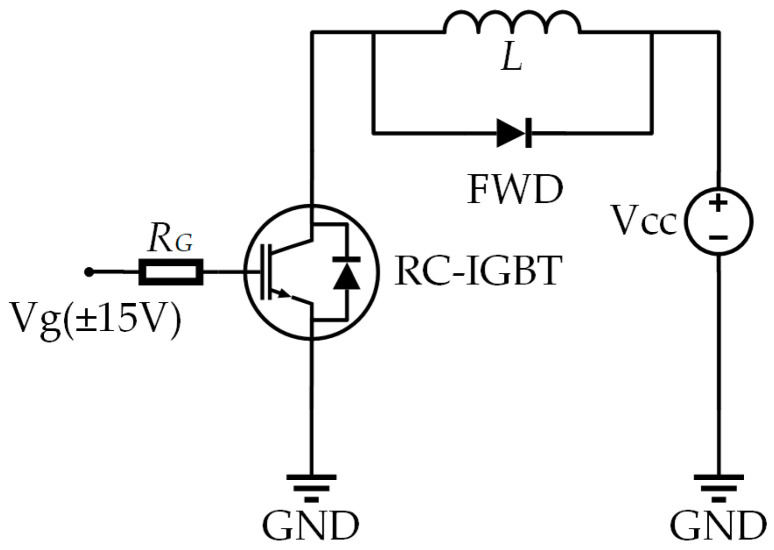
Inductive load circuit for the switching simulation of RC-IGBT.

**Figure 10 micromachines-14-00873-f010:**
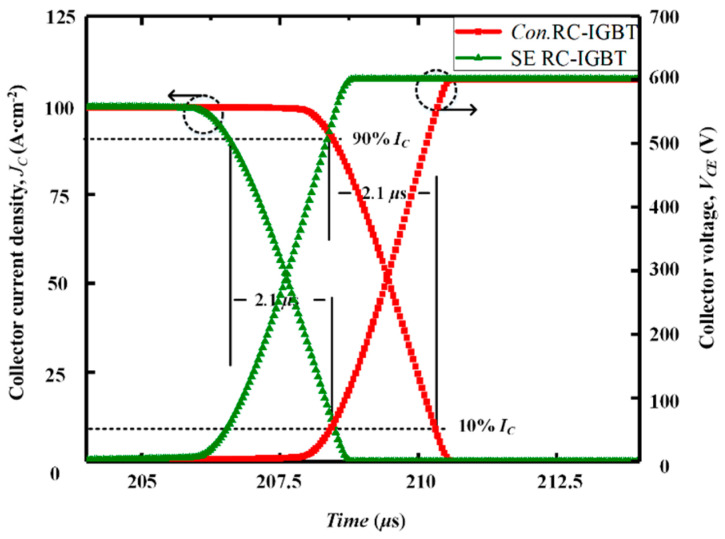
IGBT turn-off I-V characteristics of conventional RC-IGBT and SE RC-IGBT.

**Figure 11 micromachines-14-00873-f011:**
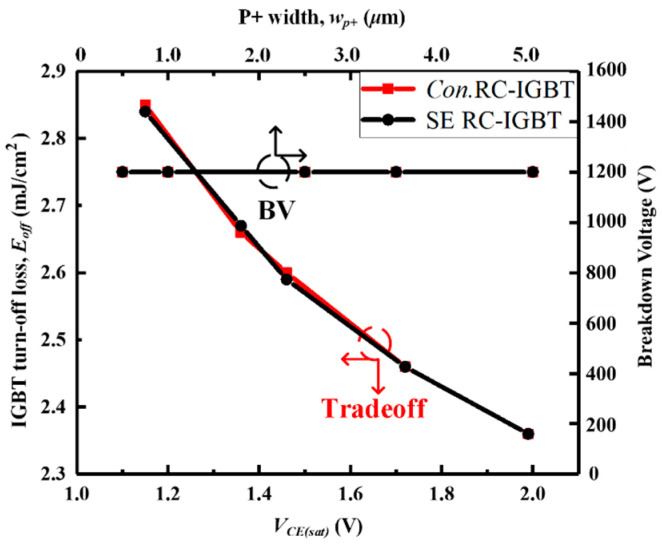
Breakdown voltage at different *w_p+_* and trade-off relationship between *E_off_* and *V_CE_*_(*sat*)_ of conventional RC-IGBT and SE RC-IGBT for different P-collector doping concentrations.

**Figure 12 micromachines-14-00873-f012:**
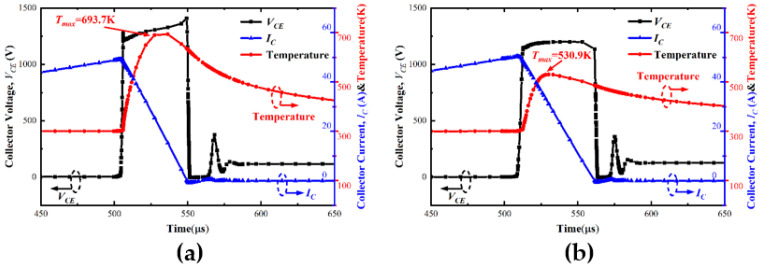
The unclamped inductive switching (UIS) waveforms for (**a**) conventional RC-IGBT and (**b**) the proposed SE RC-IGBT.

**Figure 13 micromachines-14-00873-f013:**
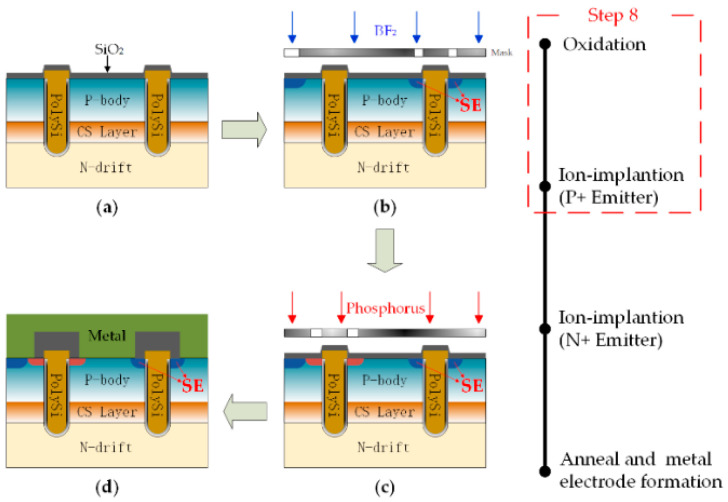
Key process steps for fabricating the front side of the SE RC-IGBT. (**a**) Oxidation; (**b**) P+ emitter ion implantion; (**c**) N+ emitter ion implantion; (**d**) Annealing and metal electrode formation.

**Figure 14 micromachines-14-00873-f014:**
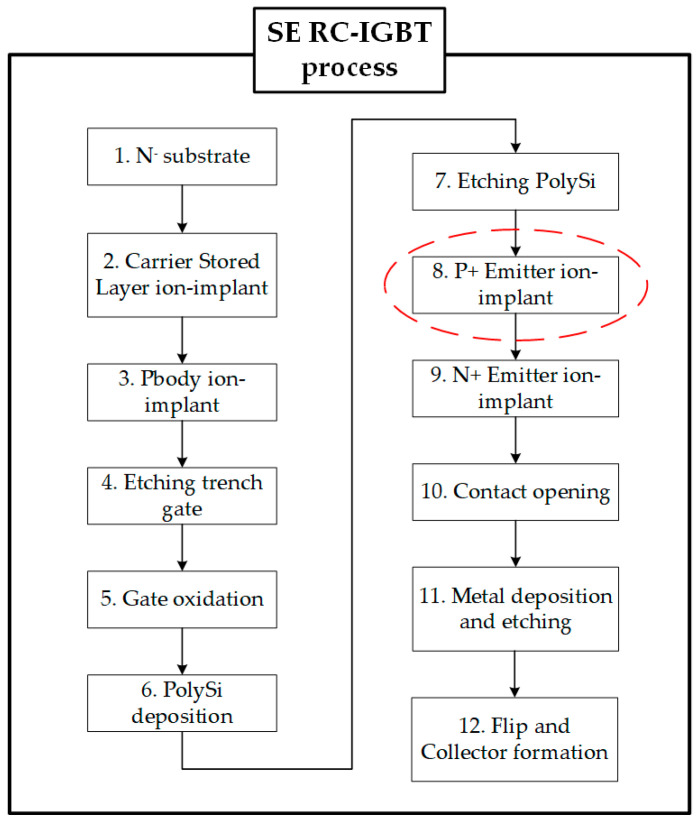
Process flow for SE RC-IGBT.

**Figure 15 micromachines-14-00873-f015:**
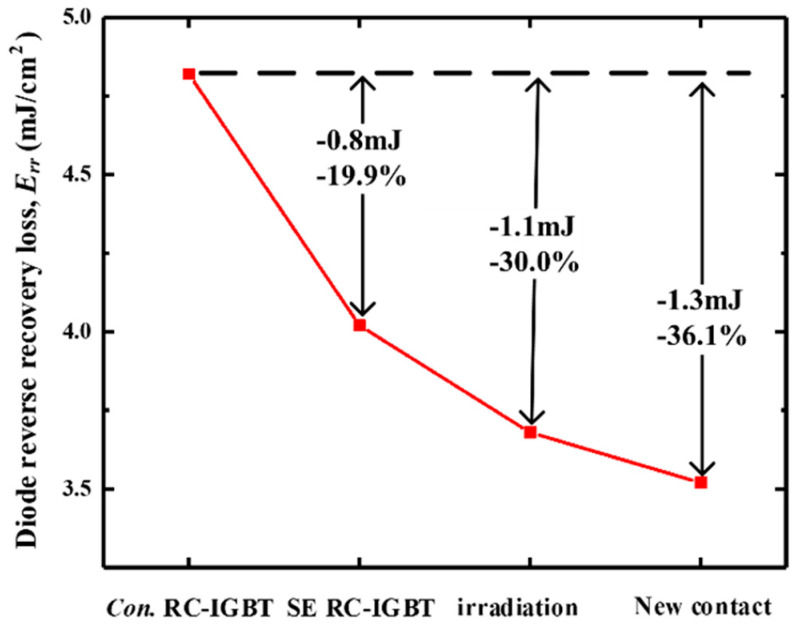
The built-in diode’s reverse recovery loss *E_rr_* for several RC-IGBT.

**Table 1 micromachines-14-00873-t001:** Device specifications.

Structure Parameters	*Con.* RC-IGBT	Proposed
P+ width (μm), *w_p+_*	5	0.5
P+ doping concentration (cm^−3^), *N_AP+_*	1 × 10^20^	1 × 10^20^
P+ thickness (μm), *t_p+_*	0.2	0.2
Wafer thickness (μm), *t_w_*	100	100
Cell width (um), *w_cell_*	270	270
P-collector width (μm), *w_Pc_*	240	240
Trench pitch (μm), *T_p_*	6	6

## Data Availability

Not applicable.

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
