# Peer review of "Low Switching Loss Built-In Diode of High-Voltage RC-IGBT with Shortened P+ Emitter"

_micromachines, 2023, doi:10.3390/mi14040873_

Round 1

Reviewer 1 Report

The authors propose a shortened P+ emitter structure to reduce the hole injection and the reverse recovery loss of the RC IGBT. The related references are well attached. As shown in fig. 3 the hole injection rate was significantly reduced only by decreasing the P+ width. However, those approaches would be able to compensate for some advantages of conventional IGBTs.

Although it will depend on the application of the IGBTs, the reduced P+ area would be able to lead to the reduced rupture voltage or UIS(unclamped inductive switching) capability. Hopefully, the authors need to compare the rupture voltages and UIS capabilities of the devices. These simulations will require thermodynamic models.

Author Response

Point 1: Although it will depend on the application of the IGBTs, the reduced P+ area would be able to lead to the reduced rupture voltage or UIS(unclamped inductive switching) capability. Hopefully, the authors need to compare the rupture voltages and UIS capabilities of the devices. These simulations will require thermodynamic models.

 Response 1: the comparison of UIS capabilities for the proposed SE RC-IGBT and the conventional RC-IGBT is shown in Fig. 12. And the related description is added from line 169 to line 179.

Reviewer 2 Report

1.       Please check the typos and grammar all over the paper. Examples: First sentence in abstract – ‘build-in’, Introduction first sentence - the Reverse-Conducting Insulated Gate Bipolar Transistor (RC-IGBT) ‘have’ been proposed.

2.     Figure captions: Sub-captions are repeated in the main caption in all the figures.

3.      J_PR should be defined when it is introduced first time in line 62.

4.     The representation of Fig. 1 and Fig. 2 are disconnected because it is not obvious for which structure of Fig. 1, the carrier distribution and turn-off characteristics in Fig. 2 are presented.

5.     Fig. 2 (b): It is very hard to understand the graph. Please explain what is presented in this plot.

6.     Table 1 parameters could be shown on Fig 1 cross sections for better understanding of lateral (width) and vertical (thickness) dimensions of various layers/regions.

7.     Fig. 6: What is the red colored curve representing?

8.     Fig. 7: The dash line to show the difference between conventional and proposed is making the figure unnecessarily crowded.

9.     Fig. 11: How are you getting different breakdown voltages? Are your changing the thickness and doping of the drift layer and calculating the turn-off loss to plot this?

Author Response

Point 1: Please check the typos and grammar all over the paper. Examples: First sentence in abstract – ‘build-in’, Introduction first sentence - the Reverse-Conducting Insulated Gate Bipolar Transistor (RC-IGBT) ‘have’ been proposed.

 Response 1: the typos and grammar all over the paper is checked. The ‘build-in’ in line 8, 89, 115, 119, 159 and 176 have been modified to ‘built-in’. The ‘have been proposed’ in line 22 has been modified to ‘has been proposed’. In addition, Some parts of the abstract and the introduction have been rewritten. And some grammatical errors have been also corrected in the paper.

Point 2: Figure captions: Sub-captions are repeated in the main caption in all the figures.

Response 2: the sub-captions in Fig.1. and in Fig.2 have been deleted.

Point 3: J_PR should be defined when it is introduced first time in line 62.

Response 3: the JPR is defined as the peak reverse recovery current in line 45.

Point 4: The representation of Fig. 1 and Fig. 2 are disconnected because it is not obvious for which structure of Fig. 1, the carrier distribution and turn-off characteristics in Fig. 2 are presented.

Response 4: the Fig. 2 show the linearized reverse recovery current density waveform of built-in diode and the change in car-rier distribution at different time intervals during the turn-off process of built-in diode. The main caption in Fig. 2 and the illustration of Fig. 2 have been rewritten in line 60 to line 68.

Point 5: Fig. 2 (b): It is very hard to understand the graph. Please explain what is presented in this plot.

Response 5: the Fig. 2 (b) has been redrawn. And the illustration of Fig. 2 have been rewritten in line 60 to line 68.

Point 6: Table 1 parameters could be shown on Fig 1 cross sections for better understanding of lateral (width) and vertical (thickness) dimensions of various layers/regions.

Response 6: the parameters in Table 1 have been labeled in Fig. 1.

Point 7: Fig. 6: What is the red colored curve representing?

Response 7: the red colored curve in Fig. 6 represents the change of collector voltage during reverse recovery of built-in diode for conventional RC-IGBT. And three collector voltage curves for SE RC-IGBT with wp+ = 0.5μm, SE RC-IGBT with wp+ = 2.5μm and conventional RC-IGBT have been added in Fig. 6. The main caption in Fig. 6 has been also rewritten.

Point 8: Fig. 7: The dash line to show the difference between conventional and proposed is making the figure unnecessarily crowded.

Response 8: the Fig. 7 has been redrawn to remove a dash line.

Point 9: Fig. 11: How are you getting different breakdown voltages? Are your changing the thickness and doping of the drift layer and calculating the turn-off loss to plot this?

Response 9: the different breakdown voltages in Fig. 11 were obained by changing wp+. And the different turn-off losses were calcaulated by changing P-collector doping concentration. The main caption in Fig. 11 has been corrected. The illustration of Fig. 11 have been also rewritten in line 162.

Round 2

Reviewer 2 Report

The revised version has been improved but there are still some question/comments which are given below.

1.     Fig-7: The original manuscript presented peak reverse recovery current but the revised one presents reverse recovery current density although the numbers are exactly same, is your area 1 cm2? If so, should have mentioned it somewhere.

2.     Fig-10: Once again, the current in the primary vertical axis has been changed to current density, how?

3.     Fig-11: I understand the point that conventional, and the proposed device are same in terms of turn-off loss and breakdown voltage. However, the scale on the secondary vertical axis (breakdown voltage) has been changed from the original manuscript (600 V to 1200 V) whereas all other parameters are the same. This is very confusing. Please explain.

4.     Fig-11: Please mention somewhere that you are getting different VCE(sat) by changing the collector doping concentration.

It is difficult to re-review a paper when parameters are changed from the original manuscript without providing a proper explanation. For example, at the beginning of section-3, it is clearly mentioned that according to Table 1, the authors are presenting data for a 1.2kV device but in the original manuscript Fig-11 represented 600V whereas the revised one represents 1.2kV. However, trench pitch is changed from 5 µm to 6 µm but I failed to find the reason. I guess, all the graphs are based on 1.2 kV device which could possibly be clarified in a better way. Another example, Fig-12 is a completely new graph, and I am not against its content but not in support of absolutely new data.

Author Response

Response to Reviewer 2 Comments

Point 1: Fig-7: The original manuscript presented peak reverse recovery current but the revised one presents reverse recovery current density although the numbers are exactly same, is your area 1 cm2? If so, should have mentioned it somewhere.

Response 1: The area of the active region in the simulated device is set to 1cm2 in this paper. After careful consideration, we have decided to use current density, which is a more rigorous parameter, to describe the current characteristics of the device. The active area has been mentioned in line 111 to 112: The area of the active region of the SE RC-IGBT is set to 1cm2.”

Point 2: Fig-10: Once again, the current in the primary vertical axis has been changed to current density, how?

Response 2: As mentioned in Response 1, the active area of the simulated device is 1cm2. So we use the current density to illustrate the current characteristics of the SE RC-IGBT.

Point 3: Fig-11: I understand the point that conventional, and the proposed device are same in terms of turn-off loss and breakdown voltage. However, the scale on the secondary vertical axis (breakdown voltage) has been changed from the original manuscript (600 V to 1200 V) whereas all other parameters are the same. This is very confusing. Please explain.

Response 3: The research is based on 1200V device in this paper. The secondary vertical axis of Figure 11 was incorrect in the original manuscrip due to our careless. So we correct this error.

Point 4: Fig-11: Please mention somewhere that you are getting different VCE(sat) by changing the collector doping concentration.

Response 4: the related discripotion “ for different P-collector doping concentration” has been added in the main caption of Figure 11.

Point 5: It is difficult to review a paper when parameters are changed from the original manuscript without providing a proper explanation. For example, at the beginning of section-3, it is clearly mentioned that according to Table 1, the authors are presenting data for a 1.2kV device but in the original manuscript Fig-11 represented 600V whereas the revised one represents 1.2kV. However, trench pitch is changed from 5 µm to 6 µm but I failed to find the reason. I guess, all the graphs are based on 1.2 kV device which could possibly be clarified in a better way. Another example, Fig-12 is a completely new graph, and I am not against its content but not in support of absolutely new data.

Response 5: We apologize for any confusion caused by our changes in the original manuscript. The reason why we changed the scale on the secondary vertical axis in Figure 11 has mentioned in response 3. The trench pitch in the revised version includes the width of the trench gate in figure 1 which was not considered in the original manuscript defined as the distance between trench gates. That is why the trench pitch is changed from 5 µm to 6 µm. And the new graph figure 12 was added based on the feedback provided by another reviewer.

Round 3

Reviewer 2 Report

Thanks for the revision. No more comments from me.